# A Concept Analysis of Illness Intrusiveness in Chronic Disease: Application of the Hybrid Model Method

**DOI:** 10.3390/ijerph19105900

**Published:** 2022-05-12

**Authors:** Youngjoo Do, Minjeong Seo

**Affiliations:** 1College of Nursing, Gyeongsang National University, Jinju 52727, Korea; suskind0523@gnu.ac.kr; 2College of Nursing, Gerontologic Health Research Center in Institute of Health Science, Gyeongsang National University, Jinju 52727, Korea

**Keywords:** illness intrusiveness, chronic disease, concept analysis, hybrid model

## Abstract

This study clarifies the concept of illness intrusiveness in patients with a chronic disease using the hybrid model method. To clarify the dimension, attributes, and definition of illness intrusiveness in chronic disease, three phases of analysis were conducted. In the theoretical phase, a working definition was devised through a systematic review. In the fieldwork phase, individual in-depth interviews were conducted with nine participants with chronic diseases. In the final analytic phase, the results were integrated through comparison and review. There are four domains and eleven attributes of illness intrusiveness in chronic disease. The domains include physical, psychological, social/contextual, and spiritual. The physical domain consists of four attributes: pain, fatigue, physical malfunction, and change of body image. The psychological domain consists of three attributes: psychological weakness, uncertainty, and stigma. The social/contextual domain is made up of three attributes: withdrawal of role play, limit of daily life, and burden of changing health habits. Finally, the spiritual domain had one attribute: unstable spiritual state. Thus, based on the study findings, it is necessary to develop a suitable illness intrusiveness in chronic disease assessment scale to assess chronic disease patients.

## 1. Introduction

### 1.1. Necessity of Study

In modern society, the prevalence of chronic diseases has been increasing due to population aging, lifestyle changes, and environmental pollution [1]. In Korea, chronic diseases account for 80.8% of all deaths; among the top ten causes of death, seven were identified to be chronic diseases [2]. Moreover, chronic diseases also present a high economic burden for patients, accounting for 83.7% of total medical expenditure in 2018, and a continued increase in this percentage is forecasted [3].

Chronic diseases refer to persistent, latent, and incurable pathological conditions that require long-term treatment and care for at least six months [4]. In other words, chronic diseases refer to diseases that cause the patients or caregivers to experience physical, psychological, social, and economic difficulties. These difficulties are due to the gradual decline in physical and physiological function while experiencing repeated cycles of improvement and exacerbation of disease. After the initial disease onset, patients experience lifelong struggles related to the disease, while self-care, including dietary control, exercise, and medication, is essential for patients with such chronic diseases [5]. The patients experience various difficulties due to lifestyle changes caused by the disease. In particular, physical limitations, dysfunction, or constrains of social relationships due to a chronic disease cause patients to experience various forms of stress, which can have a negative effect on their quality of life (QOL), including their mental health due to depression [6].

A study on the influencing factors of QOL in patients with a chronic disease by Devins [7] presented illness intrusiveness perceived by the patients as a major influencing factor. Illness intrusiveness refers to how much illness-related disruptions interfere with continued involvements in activities valued by the patient. Devins [7] used the Illness Intrusiveness Rating Scale to demonstrate that illness intrusiveness affects subjective well-being via two pathways: (1) direct effect on subjective well-being due to a decrease in positive experience following less opportunities to participate in valued activities or interests and (2) secondary negative effect on subjective well-being due to the loss of control over important parts of life.

Regarding previous global studies on perceived illness intrusiveness, a study by Renn et al. [8] on level of illness intrusiveness in patients with chronic obstructive pulmonary disease reported that one’s perception of illness intrusiveness is influenced by illness-related pain or functional limitations and treatment-related time or adverse events. A study by Mullins et al. [9] on anxiety among patients with chronic disease reported that age, illness severity, illness uncertainty, and social support influenced the differences in level of illness intrusiveness. However, despite various studies on illness intrusiveness in patients with chronic disease, from the development of the illness intrusiveness tool in 1994 to 2021, there have been no studies on the concept analysis of illness intrusiveness.

Regarding previous studies in Korea on perceived illness intrusiveness and the significant influencing factors thereof, Son’s [10] study on patients with chronic liver disease identified the significant influencing factors of role restriction and physical pain; Kim and Lee [11] identified the influencing factors of fatigue, dysfunction, marital status, and depression for patients with rheumatoid arthritis; and Kim and Roh [12], in a study on stroke patients, identified the influencing factor of functional independence. Among the domains of illness intrusiveness perceived by stroke patients, the areas most affected appeared in the order of work, health, active leisure activities, and finance [11,12].

In Korea, illness intrusiveness is also referred to by various other terms, including “illness-induced changes”, “perceived illness effect”, and “illness disability”. Illness intrusiveness is an important concept with respect to the QOL or subjective well-being of patients with chronic disease; therefore, it is necessary to use consistent terminology. For this, it is necessary to identify the attributes of perceived illness intrusiveness experienced by patients with chronic diseases. Furthermore, a review of available literature, including literature from the field of nursing, and others is necessary. The hybrid model concept analysis used by Schwartz-Barcott and Kim [13] enabled conceptualization in nursing practice by combining a theoretical phase through a literature review and a fieldwork phase that integrated the empirical contents inductively extracted from the nursing practice. This model offered the benefit of being able to identify the essential attributes of a concept at a measurable level. Accordingly, the present study applies the hybrid model concept analysis with a theoretical analysis and in-depth patient interviews. The combination of empirical contents derived for the purpose of clarifying the domains and attributes of the concept of perceived illness intrusiveness in patients with chronic diseases allows a thorough investigation into the definition of the concept.

### 1.2. Objectives

The primary objective of this study is to clarify the definition and empirical evidence of perceived illness intrusiveness in patients with chronic diseases through the hybrid model concept analysis used by Schwartz-Barcott and Kim [13]. The specific objectives are as follows:Derive a working definition of perceived illness intrusiveness in patients with chronic diseases through a review of literature;Derive the definition of perceived illness intrusiveness in patients with chronic diseases through in-depth patient interviews;Combine the comparative analysis results on data obtained through the above methods to derive the final definition of perceived illness intrusiveness in patients with chronic diseases and present the empirical evidence.

## 2. Materials and Methods

### 2.1. Study Design

This study is a methodologies study that used the hybrid model concept analysis method consisting of three phases (theoretical, fieldwork, and analysis phases) to examine the domains and attributes and confirm the definition of perceived illness intrusiveness in patients with chronic disease.

### 2.2. Procedures

#### 2.2.1. Theoretical Phase

In the theoretical phase, the domains, attributes, and a tentative definition of the concept were derived through a systematic literature review. The theoretical part of the study was a systematic review conducted in accordance with the stages proposed by the Centre for Reviews and Dissemination (CRD) guidance for conducting reviews in health care [14].

First, the review question was focused on “the essence at the core of illness intrusiveness” and “domains and attributes that can be reflected in defining the essence of the concept”.

Second, the article type was that the current study examined every original article published on the topic of illness intrusiveness, including quantitative, qualitative, mixed-method, and instrument development studies.

Finally, the search strategy was that Korean and foreign academic articles published between January 1995 and July 2021 were selected through a search using a combination of search terms. For this, databases (DBs) were divided for two researchers to perform the search. The search engines used included *PubMed*, *CINAHL*, *SCOPUS*, and *Embase* for foreign literature and *RISS*, *DBpia*, and *KCI* for Korean literature. The search terms were “chronic disease or illness” and “illness intrusiveness” in English and Korean. The inclusion criteria for the articles analysed in the present study were those including survey studies, qualitative studies, literature review studies, and reviews that examined perceived illness intrusiveness in patients with chronic diseases. Two researchers reviewed and organised the searched articles. Exclusion criterion was non english language.

The search yielded a total of 1004 articles (919 foreign and 85 Korean articles), of which 258 duplicates were excluded. Subsequently, a total of 702 articles were excluded based on a title review for the following reasons after first review. We searched additional records through hand searching key journals and trial registers. No further data could be found. A title search was used as the primary review, and 702 documents were excluded: In English, 662 documents unrelated to “illness intrusiveness”, 3 documents from the acute phase, 4 cases from paediatric disease, and 4 study designs that did not fit the topic were excluded. In Korean, 27 documents unrelated to “illness intrusiveness”, 3 documents related to the acute phase, and 27 documents unrelated to “illness intrusiveness” were excluded. In the second review, abstracts were searched, and three foreign and four Korean literatures that were not related to illness intrusiveness were excluded. The full texts of the remaining 40 articles were reviewed. Then, the Transparent Reporting of Evaluations with Non-randomised Designs (TREND) checklist was used to assess the quality of the quantitative articles and the Consolidated Criteria for Reporting Qualitative research (COREQ) checklist for qualitative articles [14]. At this stage, 3 articles were excluded, and 37 articles remained. Consequently, a total of 37 articles were included in the final analysis (Figure 1). The 37 selected articles were difficult to differentiate by field of study and therefore were divided into 6 articles on nursing and 31 articles on other fields, such as medicine and psychology.

#### 2.2.2. Fieldwork Phase

a.Data collection

In a hybrid model, the fieldwork phase involves the observation of phenomena related to a concept; furthermore, the meaning and attributes of the concept that appeared in the theoretical phase must be verifiable and in-depth interviews must be conducted appropriately [15]. In the present study, in-depth interviews were conducted individually with nine patients with chronic disease who were selected by purposeful sampling.

The participants in the study were convenience sampled by posting announcements on the bulletin boards of an apartment complex in Jinju city. For the fieldwork phase of a hybrid model, three to six individuals who can be contacted repeatedly would be appropriate [16]. Because there is no word in Korean for “illness intrusiveness”, three more people were interviewed to ensure data saturation. The participants were adults aged 19 years and older who had been diagnosed with a chronic disease (hypertension, diabetes, arthritis, cancer, etc.) and have received treatment, including drug therapy, for the chronic disease for at least one year.

Data were collected between 15 and 30 July 2021. Those who consented to participate in the study were given the option between face-to-face or telephone interviews considering the COVID-19 pandemic and other precautions. For the three participants who opted for face-to-face interviews, the interviews were conducted in an independent space (café or office). For others, the interviews were conducted over the telephone. Each individual in-depth interview was conducted for approximately 50–60 min at an agreed time. The interviews were conducted in a quiet place, ensuring that the interview could be recorded, and the participants could talk comfortably. Structured key questions and non-structured questions were asked for verifying the attributes of perceived illness intrusiveness to be analysed in the theoretical phase. These include: “Do you have difficulties due to your illness?”; “Can you tell us about changes in your daily life while having to treat and manage your diagnosed illness?”; “What is the most difficult part of living with an illness?”; “What is the most difficult part of the treatment process?”; “If you feel you are having more difficulty than others, what is the reason for this feeling?”; and “In your mind, what is the meaning of living well with a chronic disease?”

b.Data analysis and rigor

To assure the rigor of the study, validity was established by testing the credibility, transferability, dependability, and confirmability, which are the assessment criteria for qualitative studies proposed by Lincoln and Guba [16]. For credibility, the researchers attempted to build trust with the participants by conversing as much as possible and relieving tension before conducting the in-depth interviews. Moreover, the participants were instructed to read the concept analysed from the interview data to ensure that it matched their own experiences. Furthermore, it was determined that the transferability of the study was also satisfied since the results were derived through the participants expressing their own experiences of being diagnosed with a chronic disease and dealing with the disease for more than a year. To ensure dependability of the study, the researcher described in detail the entire process from the initial data collection to analytical procedures and structural and conceptual interpretation with a professional in the field. To further enhance the dependability of the study, its procedure and design was revised until an agreement was reached with the opinion of a nursing professor with experience in qualitative research. Lastly, objectivity of the data was increased by adhering to the criteria for credibility, transferability, and dependability. Meanwhile, confirmability was assured by not manipulating the stories of the participants’ experiences with any intention or purpose.

Among various Korean terms used, the term for illness intrusiveness that best reflects the characteristics of Korean culture was used to conduct the concept analysis.

c.Final analysis phase

In the final analysis phase, the results from the theoretical phase and in-depth interviews in the fieldwork phase were crosschecked. The attributes identified in the theoretical phase were reviewed to derive the final empirical evidence for perceived illness intrusiveness in patients with chronic disease.

### 2.3. Ethical Considerations

Regarding data collection from participants, the study received approval from the Institutional Review Board (GIRB-G21-Y-0035). Before starting the interviews, all participants were informed of the study’s objectives and methods: their right to withdraw from the study at any time; the use of interview contents for research purposes only; the security and confidentiality for the protection of personal information: the anonymity of personal information; and the disposal of recorded data after a set storage period. Written informed consent or verbal consent over the telephone was obtained from each participant. All participants who completed the interviews were given a gift as a token of appreciation.

### 2.4. Researcher Preparation

The first author acquired knowledge about chronic diseases through twenty years of clinical fieldwork and coursework on self-care for patients with chronic disease and empirical knowledge from encountering and caring for patients with chronic disease in the field. In addition, they learned in-depth interview methods and key points by completing courses on nursing theory development and qualitative research methodologies in graduate school. The second author has taught graduate courses on nursing theories, concept analysis, and qualitative research and has experience in various qualitative research projects through 18 years of clinical experience.

## 3. Results

### 3.1. Domains and Attributes of Perceived Illness Intrusiveness in Patients with Chronic Disease Identified in the Theoretical Phase

The term “illness intrusiveness” inherently contains an “illness” aspect and an “intrusiveness” aspect. The “illness” aspect refers to a state of not being able to function normally due to acute or persistent impairment of a part or the entirety of the mind and body, and “illnesses” can be divided into infectious and non-infectious diseases [17].

In Korean, the term “illness intrusiveness” is translated and used in various words, so integration is needed.

The literature review shows that, regardless of the field of study, illness intrusiveness in patients with chronic disease was analysed as a multidimensional concept that can be classified into physical, psychological, and social/contextual domains. The attributes analysed by specific fields, such as nursing, medicine, and psychology, are as follows:

In nursing, the physical domain of illness intrusiveness in patients with chronic disease included physical pain [18], weakness [19], decreased mobility [12,18], and sexual dysfunction [19] experienced within the illness and treatment process. The psychological domain included depression [10,20] and loss of willpower [20] caused by loss of control due to the illness and treatment process. Moreover, the social/contextual domain included difficulties associated with the experiences of the disruption of roles that the patients had played well within their family or social life [10], dietary restrictions [21], financial burdens incurred during the treatment process [12], passive leisure activities [12,18], and workplace problems [12,18].

In medicine, the attributes of the physical domain of illness intrusiveness in patients with chronic disease included being affected by the illness due to pain [12,22,23,24,25,26], fatigue [22,23], and physical dysfunction [8,9,23,24,27,28] in addition to sexual dysfunction 19,22,30,46] and sleep disturbance [29]. The psychological domain included depression [11,22,30]. In the social/contextual domain, patients with chronic diseases were found to experience various role restrictions [11,26,31,32,33], dietary restrictions [21,31,32], economic loss caused by the treatment process [26,34,35], limited leisure activities [24,36,37], and workplace problems [21,26,31,32]. Lastly, the spiritual domain included the self-concept of recognizing oneself as a patient [37,38].

In psychology, the physical domain included patients experiencing disease-related pain [39,40] and physical disability [41]. The psychological domain included mood disorders such as depression [28,39,42,43], loss of self-esteem [41,44,45,46], and uncertainty due to the characteristics of chronic disease [30,40]. The social/contextual domain included work restrictions including social role impairment [44], dietary change [40], the worsening of financial status [47,48], and limited hobbies [47,48].

By combining the attributes of illness intrusiveness in patients with chronic disease identified in each field, a working definition of illness intrusiveness in patients with chronic disease in the theoretical phase can be given as follows: “Negative changes experienced physically, psychologically, socially, and contextually within the illness and treatment process, which could threaten the well-being of patients who are unable to respond effectively to such changes” (Table 1).

### 3.2. Fieldwork Phase

A total of nine participants (four males and five females) participated in the interviews during the fieldwork phase. The average age of the participants was 63.4 years old. Moreover, eight participants were married, one participant was unmarried, and three out of nine participants were employed. With respect to the types of chronic disease, three participants had diabetes; two participants had hypertension; one had a combination of chronic rheumatoid arthritis and hypertension; two had a combination of chronic irritable bowel disease and diabetes; and one was receiving conservative treatment after completing 15 rounds of chemotherapy for gastric cancer. The duration of illness varied between 2–30 years (Table 2).

The study used the content analysis method by Elo and Kyngäs [49], which uses interview transcripts to identify and derive the relationship between the central theme concept and concept. Data obtained through the interviews were repeatedly read while underlining relevant words or phrases and adding annotations in the margins. Subsequently, 62 important statements containing key content were extracted, and 7 attributes were derived from the statements by open coding, categorization, and abstraction. Through this process, illness intrusiveness in patients with chronic disease was identified to be a dynamic process that causes negative changes in daily life, including the physical, psychological, social/contextual, and spiritual domains, after the patients experiences the illness and treatment processes. The attributes of these domains are described below (Table 3).

#### 3.2.1. Disease-Related Physical Discomfort

The participants experienced various “changes in body image” due to chronic disease, including muscle loss and change in skin colour, with darkening of the tips of the hands and feet, and damage to the peripheral nervous system, such as numbness in the hands and feet. The participants also experienced “loss of fitness” due to persistent fatigue and sweating caused by the illness as well as “sleep disturbance” that caused them to wake frequently during the night, preventing deep sleep. The patients suffered from discomfort caused by their illness as well as other forms of pain due to haemorrhoids or neuropathy caused by adverse treatment effects. The following excerpts from the interview transcripts outline this theme:


*I can’t live like this with numbness in my hands and feet… But, there’s nothing I can do except go to a bathhouse… Drugs that I take don’t work and it doesn’t come back… The doctor… Look, look!! The tips of my hands and feet are dark, right? It’s dead.*
(Participant D)


*As if my hands and feet being numb all the time wasn’t enough, I sometimes feel pain like being poked with a needle. It’s not going to get any better since it’s an adverse effect from chemotherapy, but that suffering can be more severe at times than any surgery.*
(Participant D)


*My job is that of a building cleaner. My fingers and joints are swollen and painful in the morning, so I have to soak my hands in warm water for about an hour before I can squeeze and open them. Moreover, because the fingers are already bent, holding a small tool is inconvenient.*
(Participant I)

#### 3.2.2. Psychological Suffering

As participants experienced their diagnoses of the chronic diseases and lengthy treatment processes, they faced “sorrow about the stigma of being a patient” from the words and actions of others, along with “self-blame about the disease” in believing that they did not take good care of themselves.


*One time, I was taking insulin inside the dressing room at work, and even my superior knew it was diabetes, the look that he gave me… I guess the sight of injecting insulin into the stomach looked strange.*
(Participant E)


*Getting diabetes at a young age, people would assume that I did not take good care of my body. Since then, I’ve been hiding my condition.*
(Participant A)


*I was traveling with my friends when I realised I had not brought my medicine suddenly. We had been gone for two hours by car, but we had no choice but to return to my house to pick up my medicine. *[To the friends]* I was so sorry…. I felt like a sinner.*
(Participant H)

#### 3.2.3. Decline in Value of Existence

The patients with chronic disease believed that they had become “worrisome beings” who are dependent on others, causing stress to others and even needing care from others.


*Especially my daughter worries a lot. But, when we talk on the phone, maybe she’s busy, but it seems like she doesn’t want to hear what I’m saying and seems bothered… So, I felt disappointed. I felt miserable from feeling that I’ve become a person that just causes worries..*
(Participant B)

#### 3.2.4. Relinquish Control to the Disease

The participants experienced the feelings of relinquishing control over their lives, including activities of daily living, to the illness and being “overwhelmed by the disease”. They considered the illness as “external shackles,” and believed that they were no longer the owner of their own lives as the illness had taken over.


*I hate looking at myself, not being able to eat whatever I want and having to be careful. I have to calculate each and every thing before eating, which gets frustrating at times. I feel happy when my outpatient test results are good, but if the results are bad, then I feel like it’s all over. This disease has swallowed me whole.*
(Participant A)


*The fact is that I have to live the rest of my life receiving treatment. This disease will torment me until I die. I think it’s meaningless to make future plans. It’s obvious that I won’t be able to do anything if the disease gets worse*
(Participant B)

#### 3.2.5. Social Withdrawal

The participants described their withdrawal from their lives due to their chronic disease. They had a “fear of new challenges,” that is, being afraid to start something new, and they stated that their withdrawal due to the illness and looks from others had caused them to feel like an “outsider” rather than being at the centre of their own lives. Moreover, they experienced “withdrawal from the role within the family” as the role that they had played within their family structures changed or disappeared. Furthermore, they reported experiencing “financial constraints” during the disease treatment processes. The following quotations express these attributes:


*Healthy people can do a good job no matter what work they do, but people with an illness like me must be a bit more careful. Starting new work can be intimidating.*
(Participant B)


*People used to like me… I was always at the centre of gatherings and played a leader role. Now, I feel like an outsider.*
(Participant C)


*I feel sorry about being a sick mom, and I hate the fact that my kids think of me as a sick mom *[rather than]* a regular mom.*
(Participant A)


*I used to be a very outgoing person. But after I got sick, I was in a lot of pain and didn’t have much energy, so I avoided going out on my days off. As a result, I felt pushed out and withdrawn at times, even among my friends.*
(Participant I)

#### 3.2.6. Accepting Changes to Daily Life

The patients faced various changes that they needed to accept in their daily lives as they were diagnosed with the illness and being treated. Prime examples included “refraining from food”, “escaping from past lifestyles”, and “work restrictions”. Moreover, they accepted new health habits and felt the burden of integrating them into their daily life.


*I used to eat within five minutes, but now they tell me to take over 40 min to eat, no matter what. There’s no answer… Even the doctor said that is the answer.*
(Participant D)


*I have to take various medicines prescribed by the hospital three times a day. In the middle [of the day I must take] the medicine before a meal. I also have to exercise. I lost so much muscle after my last hospitalization, so I try not to skip muscle exercises.*
(Participant F)


*People like healthy people. Who likes someone with hypertension or diabetes? They had pity on me as they asked whether I am able to work even with diabetes.*
(Participant B)


*I enjoy spicy and salty foods, but since becoming ill, the number one food I should avoid is spicy and salty foods. It can be difficult to bear at times.*
(Participant G)

#### 3.2.7. Unstable Spiritual State

Instead of a positive attitude while trying to overcome their illnesses, the participants experienced a “lack of hope” from repetitive and continuous negative experiences. At one point, they expressed having experienced the “self-perception of being a patient”.


*There’s no hope for getting better… I think only worst things will come from now… When I got hospitalised after things got bad, I had the scary thought of “will I get out alive?”*
(Participant F)


*I don’t do anything except go to the bathhouse when my hands and feet get too numb at early hours like the break of dawn. I have pity on myself because I’m a cancer patient.*
(Participant D)

The definition of illness intrusiveness in patients with chronic disease identified in the fieldwork phase is: “the experience of persistent physical discomfort, psychological suffering, and a decline in value of existence due to the chronic disease; and psychological sadness from relinquishing control to the illness, as well as the difficulty of having to accept changes in daily life and social withdrawal in an unstable spiritual state”.

### 3.3. Final Analysis Phase

In the final analysis phase, the results from the previous two phases were used to compare, review, revise, and combine the empirical results to determine the attributes of illness intrusiveness in patients with chronic diseases (Table 4).

Specifically, the final attributes of illness intrusiveness in patients with chronic disease can be divided into physical, psychological, social/contextual, and spiritual domains. Regarding the physical domain, “physical pain”, “fatigue”, “physical dysfunction”, “sleep disturbance”, and “sexual dysfunction” were identified in the theoretical phase. However, “physical dysfunction” and “sexual dysfunction” were not identified in the fieldwork phase, while “changes in body image” emerged in the fieldwork phase. In the final analysis phase, “physical pain” from the theoretical phase and “pain” from the fieldwork phase were confirmed, while “fatigue” from the theoretical phase and “loss of fitness” from the fieldwork phase were combined and named as “fatigue”. Moreover, “physical dysfunction”, “sleep disturbance”, and “sexual dysfunction” were named as “physical dysfunction”. Furthermore, “changes in body image” that emerged in the fieldwork phase was confirmed.

Regarding the psychological domain, the attributes of illness intrusiveness identified in patients with chronic disease were “depression”, “loss of self-esteem”, and “loss of willpower” in the theoretical phase and “self-blame for the diseases” and “worrisome beings” in the fieldwork phase. These attributes were renamed as “psychological weakening”. Moreover, “eternal shackles” in the fieldwork phase was combined with and renamed as “uncertainty”, which was identified in the theoretical phase. Furthermore, “sorrow about being stigmatised”, which emerged in the fieldwork phase, was revised to “stigma”.

Regarding the social/contextual domain, “escaping from past lifestyle” and “burden of accepting health habits” were identified in the fieldwork phase, while “restricted leisure activities” from the theoretical phase was not identified in the fieldwork phase. In the final analysis, “fear of challenges”, “outsider”, and “withdrawal from the role within the family”, identified in the fieldwork phase, and “role impairment” from the theoretical phase were revised and renamed as “withdrawal from role playing”. Moreover, “dietary restriction”, “financial burden”, “restricted leisure activities”, and “work restrictions” from the theoretical phase and “refraining from food” and “financial constraints” from the fieldwork phase were revised and renamed as “restrictions in daily life”. “Escaping from past lifestyle” and “burden of accepting health habits”, which emerged in the fieldwork phase, were revised and renamed as “burden of changing health habits”.

Lastly, “lack of hope” and “self-perception of being a patient”, identified in the fieldwork phase, were grouped together with “self-concept” and renamed as “unstable spiritual state”.

This study’s conceptual framework of illness intrusiveness for chronic disease by hybrid model is as follows (Figure 2): Patients with chronic diseases are affected by the disease in various ways throughout their lives, which can be classified as physical, psychological, social contextual, and spiritual. The patient responds to the disease based on the attributes affected by the disease, and if the patient responds appropriately to the disease, it leads to subjective well-being. When a patient performs ineffective coding based on this attribute, they experience discomfort, which is known as illness intrusiveness.

## 4. Discussion

In the present study, the concept of illness intrusiveness in patients with chronic disease was analysed using a hybrid model proposed by Schwartz-Barcott and Kim [13], which included theoretical, fieldwork, and final analysis phases. The concept of illness intrusiveness in patients with chronic disease was divided into four domains of physical, psychological, social/contextual, and spiritual domains, with four, three, three, and one attributes identified from each domain, respectively. The 11 attributes identified were pain, fatigue, physical dysfunction, changes in body image, psychological weakening, uncertainty, stigma, withdrawal from role playing, restrictions in daily life, burden of changing health habits, and unstable spiritual state.

A diagnosis of chronic disease can not only cause physical suffering and psychological problems as a result of a long-term struggle with the disease but can also have an effect on the overall QOL of an individual, as it affects all aspects of life [50]. In addition, the individual may become more depressed due to uncertainty and pessimistic views that disrupt important parts of their life [51]. In the present study, the diagnosis of chronic disease caused the participants to experience intrusion and disruption repeatedly and irregularly in all domains of daily life, including the above-outlined domains of illness intrusiveness. At this point, if patients respond appropriately to the disease, they will maintain a healthier daily life and experience subjective well-being; however, if they do not respond appropriately, they will experience continuous discomfort, leading to illness intrusiveness. The critical role of medical staff caring for chronic diseases is to assist patients ineffectively in responding to disease situations so that disease intrusiveness does not occur.

With respect to the physical domain, “loss of fitness” and “fatigue” were attributes of illness intrusiveness in patients with chronic disease commonly identified in both the theoretical and fieldwork phases. A study by Park and Lee [52] on fatigue among patients with chronic liver disease and a study by Son (2007) on fatigue among cancer patients report that fatigue caused negative outcomes, such as decreased vitality, loss of concentration, and decline in psychological/physical work performance, which ultimately caused the decline in QOL of patients. Such findings indicate the importance of interventions for reducing fatigue or loss of fitness in patients with chronic disease to reduce the illness intrusiveness they experience. Regarding the attribute “sexual dysfunction” identified in the theoretical phase, a study by Han and Park [53] on haemodialysis patients reported that patients have sexual dysfunction due to the disease, causing problems in maintaining and enjoying sexual activities. However, most patients do not actively seek treatment for such sexual dysfunction. In the present study, this attribute was not identified in the fieldwork phase, which may have been due to the participants avoiding specific answers to questions due to the cultural background of Korea, where discussing sex with others is difficult. It is necessary for future studies to build a more intimate and trusting rapport with the participants by increasing the duration and frequency of interviews to re-examine aspects related to sexual dysfunction. Meanwhile, “changes in body image” was an attribute in the physical domain newly identified in the fieldwork phase. Body image is the image of one’s own body developed through subjective and psychological experiences, which is affected by various factors such as an illness, surgery, or an accident. Moreover, it is an important concept that affects self-identity. A study by Cho et al. [54] on body image and adjustment to disease in children with cancer reported that adjusting well to the disease could lower body distortion and negative body image. The participants in the present study also experienced changes in body image due to loss of vascular elasticity and skin discoloration caused by peripheral nerve damage, whereby they experienced physical illness intrusiveness.

“Uncertainty” in the psychological domain was an attribute identified commonly in both the theoretical and fieldwork phases. The uncertainty of not being able to form subjective judgment during the disease and treatment processes due to a lack of information and the unstable situation of disease progression [51] caused the patients to underestimate their coping ability and form false opinions, whereby they were unable to control the series of illness-related events on their own [55]. In the fieldwork phase of the present study, the participants experienced psychological pressure from uncertainty, including not knowing when their illness will come to an end and facing anxiety about the treatment outcomes. “Sorrow about the stigma of being a patient” was also newly identified in the fieldwork phase. Social stigma causes individuals to have negative emotions such as anxiety, including stress, which leads to the formation of a negative self-concept and destruction of self-integration [56]. A study by Songtag [57] reported that patients feared social discrimination, and suffering from stereotyping is a bigger problem than physical suffering for patients who have experienced an illness.

Among the attributes in the social/contextual domain, “burden of changing health habits” was newly identified in the fieldwork phase. For patients with chronic disease, change in health habits represent a prerequisite for long-term prevention and management of illness that must be sustained not only in clinical settings, such as the hospital, but also in their daily lives. However, it is difficult to change the long-held habits in their daily lives for various reasons. Firstly, short-term behavioural changes often do not produce noticeable results, and in such situations, it can be difficult to maintain motivation and they may easily turn back to their old habits due to familiarity with the past and preferred desires. In this process, most patients experience guilt [58]. In the present study, the burden of changing health habits was identified as a factor that caused difficulties in the daily lives of patients with chronic disease. Accordingly, healthcare providers should continue to research various intervention methods for promoting continued changes in lifestyle among patients with chronic disease.

Lastly, the patients with chronic disease showed unstable spiritual states with “lack of hope” and “recognition of themselves as being stuck as a patient”. Hope helps to activate motivation to overcome adversity, reset goals, and find various solutions [59]; however, patients with chronic disease experience a lack of hope with the belief of no future for themselves due to powerlessness and hardships they have faced during the long fight with their illness and treatments. Self-stigma refers to the negative attitude that a patient has toward his or her disease and focusing on such an attitude [60]. Because patients with chronic disease have the feeling of constantly being judged and monitored by others during their treatment processes, they form a negative attitude toward themselves [61]; such self-stigma causes a decline in the motivation for self-care behaviour among patients with chronic disease, which can lead to social isolation due to difficulty in forming social relationships [62]. The participants in the present study also viewed themselves as patients and had negative views of themselves, while participants with high severity of such feeling experienced social isolation.

The limitations in the present study include the fact that in-depth interviews were conducted on only nine participants in the fieldwork phase and that there was a limited number of chronic diseases involved. Consequently, there are limitations in explaining illness intrusiveness for all chronic diseases. In particular, patients with mental illness, which have the characteristics of a chronic disease, were not included in the study. Therefore, future studies should include various chronic diseases to expand the concept of illness intrusiveness in patients with chronic disease. The attributes of illness intrusiveness in patients with chronic disease identified through the hybrid model concept analysis were derived by reflecting the negative state inherently contained in the concept of illness intrusiveness. Therefore, positive states that may develop during the treatment process were not analysed. In addition, the attribute “sexual dysfunction” could not be derived in the physical and fieldwork phases due to the limitation of interviews not providing enough opportunities to build rapport between the researchers and participants. Therefore, future qualitative study with more repeated interview sessions is recommended. Lastly, this study has limitations in that only papers written in English and Korean were used for the theoretical phase and in analysing illness intrusiveness by Koreans during the fieldwork phase. More research will be needed to determine how illness intrusiveness can be defined and what attributes are available in other cultures.

## 5. Conclusions

The present study used the hybrid model concept analysis method by Schewartz-Barcott and Kim [13] to derive the definition and attributes of illness intrusiveness in patients with chronic disease. The findings in the study showed that the concept of illness intrusiveness in patients with chronic disease can be defined as “The experience of physical intrusiveness such as pain, fatigue, physical dysfunction, and changes in body image from being diagnosed with the chronic disease and having negative experiences with the continuous, cyclic, and latent progression of the illness; psychological intrusiveness such as psychological weakening, uncertainty, and stigma; social/contextual intrusiveness such as withdrawal from role playing, restrictions in daily life, and the burden of changing health habits; and spiritual intrusiveness, which could threaten the well-being of patients who are unable to respond effectively to such changes”.

The significance of the present study is that it identified the multi-dimensionality and attributes of illness intrusiveness in patients with chronic disease to present empirical evidence for each attribute and a definition of illness intrusiveness in patients with chronic disease suitable for Korea. Furthermore, attributes of illness intrusiveness that patients with chronic diseases experience, such as “changes in body image”, “feeling like an outsider”, and “lack of hope”, were newly identified in the fieldwork stage, which also represent key significance of the present study.

## Figures and Tables

**Figure 1 ijerph-19-05900-f001:**
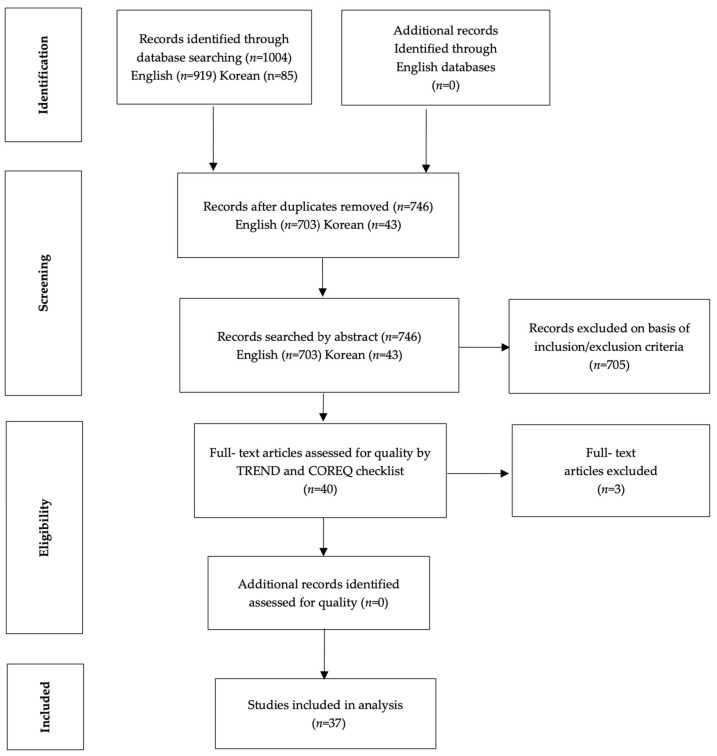
Flow diagram from literature search.

**Figure 2 ijerph-19-05900-f002:**
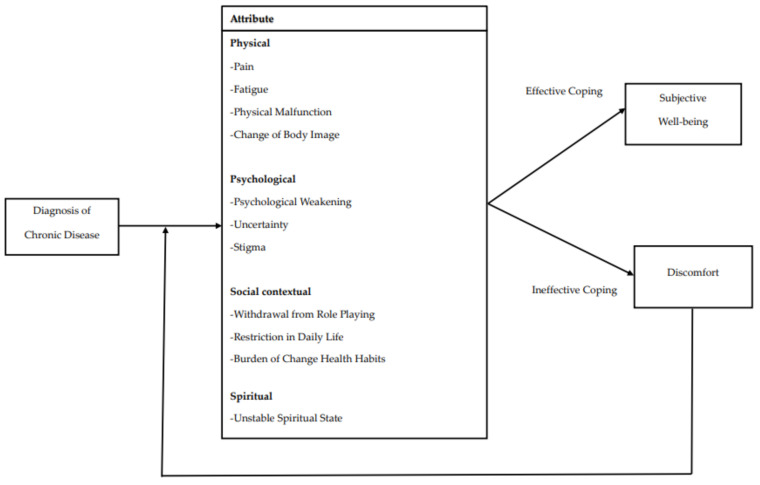
Conceptual model of illness intrusiveness for chronic disease.

**Table 1 ijerph-19-05900-t001:** Attribute of Illness Intrusiveness in Chronic Disease according to Academic Phase.

Domain	Attributes		
	Nursing	Medicine	Psychology
Physical	Physical pain [18]	Pain [12,22,23,24,25,26]	Disease-related pain [39,40]
	Weakness [19]	Fatigue [22,23]	
	Decreased mobility [12,18]	Physical dysfuction [8,9,23,24,27,28]	Physical disability [41]
	Sexual dysfuction [19]	Sexual dysfuction [18,22,26,35]	
		Sleep disturbance [29]	
Psychological	Depression [10,20]	Depression [11,22,30]	Mood disorder [28,39,42,43]
			Loss of self-esteem [41,44,45,46]
	Loss of willpower [20]		
			Uncertainty [30,40]
Social/contextual	Distruption of role play [10]	Role restriction [11,26,31,32,33]	Social role impairment [44]
	Dietary restriction [21]	Dietary restriction [21,31,32]	Dietary change [40]
	Financial burden [12]	Economic loss [26,34,35]	Worsening of financial state [47,48]
	Passive leisure activities [12,18]	Limited leisure activities [24,36,37]	Limited hobbies [47,48]
	Workplace problem [12,18]	Workplace problem [21,26,31,32]	Restriction of work [47,48]
Spiritual		Self-concept [37,38]	

**Table 2 ijerph-19-05900-t002:** General characteristics of participants.

Participant	Age	Sex	Disease	Occupation Status	Duration of Disease (Year)
A	48	Woman	Diabetes	No	5
B	53	Man	Diabetes	Yes	19
C	68	Woman	Crohn’s and diabetes	No	14
D	74	Man	Gastric cancer	No	5
E	55	Man	Diabetes	Yes	28
F	75	Woman	Irritable bowel Syndrome and diabetes	No	26
G	66	Man	Hypertension	No	10
H	68	Man	Hypertension	No	12
I	64	Woman	Rheumatoid arthritis and hypertension	Yes	17

**Table 3 ijerph-19-05900-t003:** Category and definition in the Fieldwork Phase.

Domains	Attributes	Code	Statement
Physical	Disease-related physical discomfort	Sleep disturbance	I could not sleep deeply after I get sick. (Participant A)
		Less of fitness	I get tired quickly. I only sleep on my day off. (Participant B)
		Changes in body image	I looked in the mirror and looked like a ‘smurf’. That head is the only one, that’s all. (Participant C)
		Pain	I’m stabbing with a needle because my hands and feet are numb. (Participant D)
Psychological	Psychological suffering	Self-blame about the disease	I feel guilty for not taking care of my body when I was young. (Participant A)
		Sorrow about stigma of being a patient	I guess I looked weird when I got a shot in my abdomen. (Participant E)
	Decline in value of existence	Worrisome beings	To my family, I have become a child to take care of. (Participant F)
	Relinquishing control to the disease	Overwhelmed by the disease	If the test results are bad, I think my life is ruined. (Participant A)
		Eternal shackles	Until I die, this disease will bother me. (Participant C)
Social/contextual	Social withdrawal	Withdrawal from the role within the family	It’s a burden for kids to see me as a sick mother. (Participant A)
		Fear of challenges	I’m scared of starting a new task. I won’t be good at it. (Participant B)
		Outsider	I think I became an outsider who is conscious of people around me. (Participant C)
		Financial constraints	Chronic diseases cost you money until you die. (Participant C)
	Accepting changes to daily life	Work restriction	Who would pick someone with diabetes? (Participant B)
		Escaping from past lifestyles	I ate fast for 50 years, but it’s hard to eat slowly. (Participant D)
		Burden of accepting health habits	If you take medicine and work out, the day goes by. (Participant F)
		Refraining from food	I can’t eat whatever I want. Eating out is more difficult. (Participant G)
Spiritual	Unstable spiritual state	Self-perception of being a patient	I feel shabby. Because I’m a cancer patient. (Participant D)
		Lack of hope	There’s no hope that this disease will be cured. (Participant F)

**Table 4 ijerph-19-05900-t004:** Attributes of illness intrusiveness of chronic disease according to three phases by Hybrid Model.

Domain	Attributes		
	Theoretical Phase	Fieldwork Phase	Final Analytic Phase
Physical	Physical pain	Pain	Pain
	Fatigue	Less of fitness	Fatigue
	Physical dysfuction		Physical malfunction
	Sleep disturbance	Sleep disturbance	
	Sexual dysfuction		
		Change of body image	Change of body image
Psychological	Depression		Psychological weakening
	Loss of self-esteem	Self-blame about the disease Worrisome beings	
	Loss of willpower	Overwhelmed by the disease	
	Uncertainty	Eternal shackles	Uncertainty
		Sorrow about stigma of being a patient	Stigma
Social/contextual	Role impairment	Fear of challenges Outsider Withdrawal from the role within the family	Withdrawal from role playing
	Dietary restriction	Refraining from food	Restrictions in daily life
	Financial burden	Financial constraints	
	Restricted leisure activities		
	Workplace problem	Work restriction	
		Escaping from past lifestyles Burden of accepting health habits	Burden of changing health habits
Spiritual	Self-concept	Lack of hope	Unstable spiritual state
		Self concept as a patient	

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
