# Peer review of "A Concept Analysis of Illness Intrusiveness in Chronic Disease: Application of the Hybrid Model Method"

_ijerph, 2022, doi:10.3390/ijerph19105900_

Round 1
Reviewer 1 Report
Line 18: Add “of” in …made up “of” three...
Line 19: misspelling role paly should be roleplay or role play
Line 23: Replace semicolons with commas in Keywords: segment
Line 27: Remove comma
Line 40: Replace comma after “...chronic diseases [5]…" with period
*Line 42: “...or restrictions in and disconnection of” rewording needed for clarity
Line 58: Add coma before and
Line 78: Remove comma
Line 197: Replace comma after dependability with a period
Line 212: Replace semicolon with colon
Line 213-215: Replace semicolons with commas
Line 306: Replace semi colon after “muscle loss” with a comma
Line 386: take away capitalization in “Self-perception”
Line 438: in the first statement: space after period, comma after “only one” in “That head is the only one that’s all.”
Line 439: No period after “shabby”
Line 471 & 474: uses lastly twice
Line 492: Table 3: Under physical attributes disturbance is misspelled twice as disterbance
Line 503: add colon after “were”
Lines 477-485 & 601-608 Repeat the final definition, consider taking out the first instance
Line 612: remove “and” after “furthermore”
Reword line 42 for clarity and consider leaving the final definition in lines 477-485 & 601-608, in just the conclusion.
Author Response
We wish to thank you for the kind reviews and valuable comments on our manuscript entitled, “A Concept Analysis of Illness Intrusiveness in Chronic Disease: Application of the Hybrid Model Method” (Manuscript ID: ijerph-1630725).
In response to the reviewers’ comments, we have made several changes to the manuscript and provided more detailed information. Especially, for literature review, Embase was added to the electronic databases, and in Korea, a search was conducted with KCI instead of Kmbase. The search strategy used CRD guidance's TREND and COREQ checklist.
This study concept development in the hybrid model by Schwartz-Barcott, D.& Kim, H.S. Field research is the step in conceptual analysis that involves verifying the content of the literature and looking for new phenomena. Schwartz-Barcott, D.& Kim recommended interviewing three to six people. We analyzed the interview of a total of 9 people by adding 3 participants based on the reviewer's recommendations. Because there is no word in Korean for "illness intrusiveness," three more people were interviewed to ensure data saturation.
We have prepared a response table that shows our responses or changes in response to comments made by the reviewers. We hope that this revision has resolved the points raised by the reviewers, and thus strengthened our manuscript.
Thank you for your consideration of this manuscript. We are especially grateful for your thorough advice.
Reviewer: 1 |
|
|
Add “of “ in .. made “of” three |
Thank you for your recommendation. done |
Line 19 |
Misspelling role paly should be roleplay or role play |
Thank you for your recommendation. Role play |
Line 19 |
Replace semicolons with commas in keywords:segment |
Thank you for your recommendation done |
Line 23 |
Remove comma |
Thank you for your recommendation done |
Line 27 |
Replace comma after “..chronic diseases (5)…” with period |
Thank you for your recommendation done |
Line 40 |
or restrictions in and disconnection of rewording needed for clarity |
Thank you for your recommendation We reworded constrains. |
Line 42 |
Add coma before and |
Thank you for your recommendation done |
Line 58 |
Remove comma |
Thank you for your recommendation done |
Line 78 |
Replace comma after dependability with a period |
Thank you for your recommendation done |
Line 220 |
Replace semicolons with commas |
Thank you for your recommendation done |
Line 235-238 |
Replace semi colon after “muscle loss” with a comma |
Thank you for your recommendation done |
Line 342 |
In the first statement : space after period, comma after “only one” in “that head is the only one that’s all.” |
Thank you for your recommendation done |
Table 3 |
No period after “shabby” |
Thank you for your recommendation done |
Table 3 |
Uses lastly twice |
Thank you for your recommendation Remove first one. |
Line 521 |
Under physical attributes disturbance is misspelled twice as disterbance |
Thank you for your recommendation done |
Changed Table 4 |
Add colon after “were” |
Thank you for your recommendation done |
Line 554 |
Repeat the final definition consider taking out the first instance |
Thank you for your recommendation done |
Line 527-535 |
Remove “and” after “furthermore” |
Thank you for your recommendation done |
Line 672 |
Reword line 42 for clarity and consider leaving the final definition or in just the conclusion. |
Thank you for your recommendation We expressed another word that was analyzed integrally academic phase and fieldwork phase. We used the word as attribute of social contextual “financial constraints”. |
Table 4 |
Reviewer 2 Report
In my opinion, 6 in-depth interviews are not enough to make a conclusion.Why did the authors conduct only 6 interviews?
What are the study selection criteria during meta-analysis?
Were publications based on quantitative or qualitative research considered? Did the quantitative studies meet criteria such as randomization? This is not explained.
Author Response
We wish to thank you for the kind reviews and valuable comments on our manuscript entitled, “A Concept Analysis of Illness Intrusiveness in Chronic Disease: Application of the Hybrid Model Method” (Manuscript ID: ijerph-1630725).
In response to the reviewers’ comments, we have made several changes to the manuscript and provided more detailed information. Especially, for literature review, Embase was added to the electronic databases, and in Korea, a search was conducted with KCI instead of Kmbase. The search strategy used CRD guidance's TREND and COREQ checklist.
This study concept development in the hybrid model by Schwartz-Barcott, D.& Kim, H.S. Field research is the step in conceptual analysis that involves verifying the content of the literature and looking for new phenomena. Schwartz-Barcott, D.& Kim recommended interviewing three to six people. We analyzed the interview of a total of 9 people by adding 3 participants based on the reviewer's recommendations. Because there is no word in Korean for "illness intrusiveness," three more people were interviewed to ensure data saturation.
We have prepared a response table that shows our responses or changes in response to comments made by the reviewers. We hope that this revision has resolved the points raised by the reviewers, and thus strengthened our manuscript.
Thank you for your consideration of this manuscript. We are especially grateful for your thorough advice.
Reviewer: 2 |
|
|
In my opinion, 6 in-depth interviews are not enough to make a conclusion. Why did the authors conduct only 6 interviews? |
Thank you for your recommendation. The comment has been revised as follows: This study concept development in hybrid model by Schwartz-Barcott, D.& Kim, H.S. Field research is the step in conceptual analysis that involves verifying the content of the literature and looking for new phenomena. Schwartz-Barcott, D.& Kim recommended interviewing three to six people. We analysed the interview of a total of 9 people by adding 3 participants based on the reviewer's recommendations. |
Line 317-325 & additional Table 2 |
What are the study selection criteria during meta-analysis?
|
We agree with your comment. We selected the original articles published on the topic of illness intrusiveness, including quantitative, qualitative, mixed method, and instrument development studies. We described the type of articles in the text. For literature review, Embase was added to the electronic databases, and in Korea, a search was conducted with KCI instead of Kmbase. Also, the search strategy used CRD guidance's TREND and COREQ checklist. As a result, one literature was added compared to the last time, and the literature analysis result table is the same except for sexual dysfuction. |
Line 110-150 & Changed Figure 1. |
Were publications based on quantitative or qualitative research considered? Did the quantitative studies meet criteria such as randomization? This is not explained. |
Thank you for your recommendation. The comment has been revised as follows: This study is a methodologies study which used the hybrid model concept development (theoretical, fieldwork, and analysis phases) to examine the domains and attributes and confirm the definition of perceived illness intrusiveness in patients with chronic disease. This study is neither qualitative nor quantitative. We selected the original articles published on the topic of illness intrusiveness, including quantitative, qualitative, meta-analysis, mixed method, and instrument development studies. We described the type of articles in the text.
|
Line 104-119 & Changed Figure 1 |
Reviewer 3 Report
This study addresses the conceptual clarification of illness intrusiveness in patients with chronic diseases by combining two methods: literature review and in-depth interviews. This is a relevant topic in general and also for IJERPH.
My main concern about the manuscript is methodological. In my opinion, neither method was performed properly.
First, I think that it is not possible to support a representative concept of illness intrusiveness without doing a systematic review. If this review was systematic, the authors would have to report it using high-quality criteria such as PRISMA-2020.
Second, as the authors state in the limitations paragraph, only six in-depth interviews is clearly insufficient to be representative of the chronic disease population.
Author Response
We wish to thank you for the kind reviews and valuable comments on our manuscript entitled, “A Concept Analysis of Illness Intrusiveness in Chronic Disease: Application of the Hybrid Model Method” (Manuscript ID: ijerph-1630725).
In response to the reviewers’ comments, we have made several changes to the manuscript and provided more detailed information. Especially, for literature review, Embase was added to the electronic databases, and in Korea, a search was conducted with KCI instead of Kmbase. The search strategy used CRD guidance's TREND and COREQ checklist.
This study concept development in the hybrid model by Schwartz-Barcott, D.& Kim, H.S. Field research is the step in conceptual analysis that involves verifying the content of the literature and looking for new phenomena. Schwartz-Barcott, D.& Kim recommended interviewing three to six people. We analyzed the interview of a total of 9 people by adding 3 participants based on the reviewer's recommendations. Because there is no word in Korean for "illness intrusiveness," three more people were interviewed to ensure data saturation.
We have prepared a response table that shows our responses or changes in response to comments made by the reviewers. We hope that this revision has resolved the points raised by the reviewers, and thus strengthened our manuscript.
Thank you for your consideration of this manuscript. We are especially grateful for your thorough advice.
Reviewer: 3 |
|
|
This study addresses the conceptual clarification of illness intrusiveness in patients with chronic diseases by combining methods: literature review and in-depth interviews. This is a relevant topic in general and also for IJERPH.
First, I think that it is not possible to support a representative concept of illness intrusiveness without doing a systemic review. If this review was systematic, the authors would have to report it using high-quality criteria such as PRISMA-2020. Second, as the authors state in the limitations paragraph, only six in-depth interviews is clearly insufficient to be representative of the chronic disease population. |
Thank you for your recommendation We search the site http://www.prisma-statement.org/. and we restated our search strategy in this article. The comment has been revised as follows This study is a methodologies study which used the hybrid model concept development (theoretical, fieldwork, and analysis phases) to examine the domains and attributes and confirm the definition of perceived illness intrusiveness in patients with chronic disease. We added the Transparent Reporting of Evaluations with Non-randomized Designs (TREND) check-list to assess the quality of the quantitative articles and the Consolidated criteria for Reporting Qualitative research (COREQ) checklist for qualitative articles. |
Line 110-152 & Changed Figure 1. |
Reviewer 4 Report
The authors of this work apply concept of illness intrusiveness in patients with a chronic disease using the hybrid model method in South Korea.
The research is clearly presented although the positioning the references of the appendix induces some confusion.
The authors could expand the discussion section so as to include some international comparison, if available, to asses the specificity of the South Korean context.
Kind regards
Author Response
We wish to thank you for the kind reviews and valuable comments on our manuscript entitled, “A Concept Analysis of Illness Intrusiveness in Chronic Disease: Application of the Hybrid Model Method” (Manuscript ID: ijerph-1630725).
In response to the reviewers’ comments, we have made several changes to the manuscript and provided more detailed information. Especially, for literature review, Embase was added to the electronic databases, and in Korea, a search was conducted with KCI instead of Kmbase. The search strategy used CRD guidance's TREND and COREQ checklist.
This study concept development in the hybrid model by Schwartz-Barcott, D.& Kim, H.S. Field research is the step in conceptual analysis that involves verifying the content of the literature and looking for new phenomena. Schwartz-Barcott, D.& Kim recommended interviewing three to six people. We analyzed the interview of a total of 9 people by adding 3 participants based on the reviewer's recommendations. Because there is no word in Korean for "illness intrusiveness," three more people were interviewed to ensure data saturation.
We have prepared a response table that shows our responses or changes in response to comments made by the reviewers. We hope that this revision has resolved the points raised by the reviewers, and thus strengthened our manuscript.
Thank you for your consideration of this manuscript.
Reviewer: 4 |
|
|
The authors of this work apply concept of illness intrusiveness in patients with a chronic disease using the hybrid model method in South Korea.
The research is clearly presented although the positioning the references of the appendix induces some confusion.
The authors could expand the discussion section so as to include some international comparison, if available, to assess the specificity of the south Korean context. |
We agree with your comment. The comment has been revised as follows: Lastly, this study has limitations in that only papers written in English and Korean were used for the theoretical phase and in analysing illness intrusiveness by Koreans during the fieldwork phase. More research will be needed to determine how illness intrusiveness can be defined and what attributes are available in other cultures. |
Line 653~656 |
Round 2
Reviewer 3 Report
The authors have satisfactorily addressed all my concerns and as a result the manuscript has been significantly improved.